# Head-Starting and Conservation of Endangered Timber Rattlesnakes (*Crotalus horridus horridus*) at Roger Williams Park Zoo

**Gabriel Montague**

Roger Williams Park Zoo, Providence, RI 02907, USA; gmontague@rwpzoo.org

**Abstract:** The timber rattlesnake (*Crotalus horridus horridus*) is extinct in Rhode Island and Maine with populations in the remaining New England states afforded endangered species status. Lou Perrotti, Director of Conservation and Research at Roger Williams Park Zoo (RWPZ), has long been a champion of these unloved animals in peril and spearheaded a program at the zoo in fall 2010 to work on the recovery of this endangered snake species. Partnering with multiple state agencies was required to begin saving this maligned native species, which had seen massive population reductions. The program began with accepting adults of varying size and sex suffering from skin lesions believed to be Snake Fungal Disease from multiple New England states. Depending on the severity of the infection, the animals were treated and then released. As the program evolved, it became a goal to not only treat affected adults and determine the overall health of declining New England populations but to begin a head-start program with one of the state conservation partners. Head-starting refers to when neonates are either born in a zoological facility or captured in the wild and raised under managed care until a desired size is reached. They are then released back to the wild, giving them a better chance for survival. The area where the snakes are kept at the zoo allows for temperature manipulation to simulate a natural temperature change and allow for the brumation of individuals. Once the appropriate size is reached, a radio transmitter is surgically implanted to allow radio telemetry tracking after release. The head-starting strategy has been a success, with individuals being found years later, suggesting they are surviving in the wild. Due to the sensitive nature of this program, some data and names of partners have been purposely omitted.

**Keywords:** rattlesnake; head-start; Conservation; zoo; endangered species

## 1. Introduction

The timber rattlesnake (*Crotalus horridus horridus*) is an iconic American animal appearing on the famous Gadsden Flag [1] and other patriotic symbols throughout the history of the United States. Despite being a patriotic symbol for the United States of America, the animal has been mercilessly persecuted by humans for being a venomous snake. Timber rattlesnakes are relatively large snakes, ranging from 35 to 60 inches in length. They are a pit viper (family Viperidae, subfamily Crotalinae) with a prominent tail rattle and heavily keeled scales. Hatred and ignorance have had a devastating effect on the timber rattlesnakes' population. New England is at the northern extant of the timber rattlesnakes' native range, so naturally, life in this region is a little more difficult due to the northern climate, even without any additional external pressures. The past two hundred years have seen bounties for dead animals, extreme habitat destruction and degradation, road mortality, and the recent emergence of Snake Fungal Disease (SFD). SFD is caused by *Ophidiomyces ophiodiicola*, a fungus that was recently split from a group of fungi long referred to as the *Chrysosporium* anamorph of *Nannizziopsis vriesii* species complex (CANV). A primary infection by *O. ophiodiicola* may initiate a series of events that lead to the death of the host snake. It is also plausible that in some instances, the fungal colonization of tissues occurs because the host's health is already compromised [2].

Timber rattlesnakes have very specific habitat requirements, with pine and mixed deciduous forests nearby to mountainous areas and access to deep underground dens to allow for brumation (a period of dormancy over the winter months). The underground dens are usually near a southern facing talus (loose rock) slope with little vegetative cover. Other specific habitat requirements include birthing areas (rattlesnakes give birth to live young) usually protected by rock ledges, basking areas, and even trees to climb [3]. Unlike true hibernation, they have periods of activity in the brumation period. Entering their dens in mid to late October, they occasionally move around in the dens and emerge in April, basking frequently until leaving to forage for the summer months. In addition, long time spans are required to attain sexual maturity, long inter-birthing intervals occur in individual females, and a small litter size make this rattlesnake vulnerable to a plethora of threats [4]. Juvenile rattlesnakes face many difficulties as they grow, as they are food for many predators including Virginia opossums, raptors, bobcats, and even other snakes. During a previous study with timber rattlesnakes, it was determined that for snakes in the first two age groups (ages 1 yr and 2–4 yr), survival tended to decline over time for both color variations (darker scalation or more yellowish), while for adult snakes (5 yr and older), survival was static or even slightly improved when raised in managed care [5].

The basic goal of the SFD study was to take in animals suffering from various stages of perceived SFD infections and, with various treatments, support the animals in achieving sufficient health so they could be released back to their home range. For the head-starting program, the neonates were raised in managed care until they attained a weight heavy enough (~300 g) to allow for a small radio transmitter to be surgically implanted. This process usually takes two and a half years of being in managed care and this size also makes it more likely the animals will survive into adulthood as they are larger than a wild counterpart of the same age.

## 2. Materials and Methods

### 2.1. Rehabilitation and Treatment of SFD

In the beginning of the project (October 2010 until 2016), the main goal was the determination of the overall health of animals suffering from skin lesions believed to be caused by SFD. To begin the rehabilitation and treatment of adult individuals suffering from severe fungal infections, a suitable space was identified for its ability to be temperature controlled with air conditioning in the warmer months and with heat in the winter months. One incredibly important detail of this room was the ability to drop the temperature and photoperiod low enough in the late fall and winter to simulate what the snakes would experience in the wild as they entered brumation. A team of keepers and veterinary technicians capable of working with venomous snakes was identified by the Director of Conservation and Lead Keeper. Since SFD is a highly contagious pathogen, a basement room was chosen since it was not close to any of the zoo's collection snakes. This room was treated as a quarantined space. A footbath containing disinfectant was used before entering and leaving the room. This served the dual purpose of limiting exposure of SFD through the tracking of fungus on footwear to the zoo's snake collection. All servicing of the zoo's collection snakes was completed before entering the quarantine area and hands were always washed prior to and after doing any work with the conservation snakes.

### 2.1.1. Husbandry

For each snake, Vision brand caging (models 211 and 322) was set up along with a water bowl, small heating pad on rheostat, appropriately sized artificial hide, fluorescent lighting (on a timer), and newspaper as a substrate. The paper was chosen over a more natural substrate for its ease of cleaning, which was very important for the animals that required SFD treatment. Keepers always worked in teams of two. This was to ensure safety and, in the event of a keeper being bitten by a rattlesnake, the snake bite protocol could be immediately initiated. In both mid spring and late fall, temperature and light cycles were manipulated to simulate the seasonal changes they would encounter in the wild. In

the late fall, roughly two weeks after the last feed, the heat pads were unplugged, and the room temperature was dropped by 5 °F every five days until the ambient temperature was 47–52 °F. The light cycle would be reduced on the days the temperature was dropped beginning with a summer cycle of 14 h light and 10 h dark. When the last cycle of 8 h light and 16 h dark was achieved, the next step was to turn the lights off completely, only turning on a room light to perform daily checks while the animals were in hibernation. This process was typically reversed beginning in late March; although, there were years when warm spring temperatures forced us to begin the process earlier. Once the snakes had been at an ambient temperature of 75–78 °F with a full photoperiod for a week, they were offered their first meal. Throughout their captivity, the snakes were offered an appropriately sized food item, usually a small rat or large mouse, and fed on a bimonthly schedule. A few of the snakes with severe cases of SFD required surgery under anesthesia for debriding of the affected area. One individual needed to recover from a perceived predator attack that it survived in the wild.

### 2.1.2. Medical Assessment and Treatment

To test for the prevalence of SFD, biopsies were taken from the scales of snakes presenting symptoms of a fungal dermatitis, and blood samples were taken to help determine the overall health of affected individuals. For the actual treatment of the rattlesnakes, the Veterinary Department determined what medical treatment was needed besides antifungal (Voriconazole) drugs. Often, secondary infections were present in the animals and the antibiotic ceftazidime along with subcutaneous fluids were used to treat these infections. When the rattlesnakes needed to be handled for a medical treatment or checkup, a keeper removed the snake from its habitat with an appropriately sized snake hook and placed the animal on the floor along a wall. The keeper would then attempt to coax the snake into an appropriately sized transparent plastic tube. Making sure the proper tube was selected was a very important part of this process. If the tube was too large, the snake could turn around and envenomate the keeper. If it were too small, the snake could become stuck and be difficult to safely remove. The keepers often had to switch the hook and tube between hands while attempting to coax the snake into the tube. The snakes, being wild, were often reluctant to take part in this exercise. When roughly 35% of the snake was inside the tube, the keeper would make a single swift movement of setting down the hook and grabbing the snake right where the tube and snake met. Once the snake had been secured in the tube, its body could be picked up with the keeper's free hand so that it was fully supported. After the required treatment was given, the snake was returned to its habitat. Usually by this point, the snake's mood had soured due to being handled and treated so it was important to be as safe as possible. The snake would, therefore, be placed in its habitat while still secured in the tube. In one fluid motion, the keeper would release the snake and tube while simultaneously using the hand holding the upper part of the tube to push the snake away from the front of the enclosure. This motion, coupled with the snake's natural inclination to back out of the tube, allowed the second keeper to close the door as the main keeper backed up.

### 2.2. Head-Starting Neonates for Release to the Wild

The project shifted from treating infected adults to head-starting in 2016. The husbandry processes remained the same (temperature and lighting), but the caging had to be quite different due to the snakes' small size (usually between 18 g–25 g). The neonate snakes were kept in rack style caging (individual tubs had measurements of 36″L × 16″W × 12″T) with a locking lid, and these contained a small water bowl and an artificial hide. A large fluorescent tube light was placed in front of the whole rack of caging.

The neonates were acquired in two separate ways. In the first, state biologists would catch a known gravid female who would give birth at the zoo. In the second, the neonates were captured in the wild at known birthing sites. Neonates were weighed, placed in their rack, and after their first shed (usually within a week), they were offered their first

meal. Although the adults were fed every other week to every third week, depending on body condition, neonates were offered food once a week. Most neonates readily accepted thawed fuzzy mice as their first meal, with a few exceptions needing to be started on thawed pinky mice. The juvenile snakes were kept until they reached a size of around 300 g when they could be surgically implanted with a small radio transmitter allowing biologists to track them when they were released. The growth process usually took around two and a half years, and they were released where they originally came from in mid to late spring depending on the weather. All snakes were also implanted with a Passive Integrated Transponder (PIT) so that in the event of the radio transmitter failure they could still be identified.

## 3. Results

Between 2010 and 2016, 96.1% of adult rattlesnakes brought in for the study and treatment of SFD were returned to the wild after being cleared of medical issues. After release, the animals were monitored by state biologists and seemed to do well, being observed year after year; however, the fungus was a continuing threat since it is present in their environment. The 3.9% of adults that were not able to be released succumbed to the effects of severe SFD and secondary infections. The head-start program that began in 2011, and is ongoing, has a success rate of 91.6% of individuals being released back to their wild habitat. The 8.4% of juveniles that were not released include the animals currently in managed care and the small percentage of animals lost due to a diagnosis of failure to thrive. Through radio tracking, it has been noted that released animals are currently thriving and successfully hibernating, which can be one of the biggest hurdles for an animal raised under human care. However, to be successful, released animals must demonstrate competency in the wild such that they can select appropriate resources, respond correctly to seasonal environmental cues, avoid predators, and ultimately grow, survive, and reproduce [6]. Due to the sensitive nature of this information, some data and locations have been purposely omitted.

## 4. Discussion

While the timber rattlesnake faces many threats from both natural and anthropogenic sources, rehabilitation and head-starting programs such as this can continue to provide some relief to struggling wild populations. The juvenile snakes, when released, are larger than a wild counterpart of the same age and the threats they face would be the same as an adult snake. Timber rattlesnakes are a vital part of the ecosystem in New England, being predators of small mammals that are primary vectors for Lyme disease. In a study conducted in the George Washington National Forest, it was observed that 87% of snakes had consumed small mammals and 13% had consumed birds (Uhler et al., 1939) [7]. They also are prey for many other native species such as bobcat, red tailed hawk, Virginia opossum and fox to name a few. Sadly, however, their value to the ecosystem is often overlooked and they are not beloved. They are victims of society's perpetuated intolerance and commonly encouraged phobias for serpents, especially venomous ones. One of the most important things timber rattlesnakes have going for them is secrecy, as very few people realize they even live here, and fewer are lucky enough to see them in the wild. There are many possible directions for future research regarding head-starting rattlesnakes, including reproductive studies of head-started snakes compared to wild snakes, long term studies of growth rates, population stability in head-started populations compared to wild snakes and its efficacy in helping to maintain and even bolster declining wild populations. Natural mortality rates for timber rattlesnakes in their first two years of life are as high as 66%, due to failure to procure food, predation, and failure to find proper hibernacula [7]. For a species that is barely clinging to life in New England, the importance of a program such as the one here at RWPZ cannot be overstated.

**Funding:** Funding for the SFD study was provided by a USFWS Regional Conservation Needs grant (RCN) number 2012-02 and grant amount was not to exceed $81,148.71. The funding for the head-start program, treatment and husbandry of all snakes is provided in house by RWPZ.

**Institutional Review Board Statement:** The animal study protocol was approved by veterinary department and senior management at Roger Williams Park Zoo.

**Informed Consent Statement:** Not applicable.

**Acknowledgments:** The author would like to thank the following: Lou Perrotti, Director of Conservation and Research at Roger Williams Park Zoo and Carousel Village and his vision and drive to work with a controversial species; the multiple New England state conservation departments that are willing to protect a species that suffers from such poor public perception, which can hamper the conservation process; my wonderful wife for taking the time to edit the review; and lastly, I would like to thank Karin Schwartz for being patient and her endless professional assistance in writing this paper.

**Conflicts of Interest:** The authors declare no conflict of interest.

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
