# Peer review of "Head-Starting and Conservation of Endangered Timber Rattlesnakes (Crotalus horridus horridus) at Roger Williams Park Zoo"

_2673-5636, doi:10.3390/jzbg3040043_

Round 1

Reviewer 1 Report

This paper reports a very interesting work, that tell us how to save an endangered unloved animals like the timber rattlesnake, what is head-starting, how to treat SFD,it has an important role to protect this species. It also let people have a correct cognition to the timber rattlesnake and get more knowledge related to the snake. It would be better if the description was more details. Despite the paper is short and simple, it has an obvious value, moreover this study is not easy work fulling with danger. I suggest editor can consider to publish it.

Author Response

The additions that have been made address some of the lack of detail 

Reviewer 2 Report

The paper looks fine, however I have suggestions and are following   

1.      It would be better if the beginning part of the second paragraph of the introduction were moved to the starting paragraph.

2.      It is better to write the methodology part more scientifically than to write it literary.

3. Are the definition and explanation for Brumation and some portion regarding the methodology necessary in the introduction part?

4.   The specification of materials used are missing.

5. Sorting the methodology part under some sub-heading will make reading easier and more interesting.

6. Please, Don’t forget to write the units of physical quantities like temperature whenever they are mentioned.

7.  In the result, only the success rate is mentioned, the reason for failure and the ways to solve it are not discussed. 

8.      The discussion part seems bit weak and it lacks some references.

Author Response

  1. This has been moved to where it makes more sense
  2. This has been modified to make it sound more scientific
  3. I believe including it gives the readers a sense of an important part of rattlesnake behavior
  4. model numbers to caging have been added for detail
  5. Subheadings have been added
  6. Temperature units have been corrected
  7. The failure rates were added in broad terms
  8. some references were added and the discussion section had some additions to give it more weight